# Susceptibility of the Cigarette Beetle *Lasioderma serricorne* (Fabricius) to Phosphine, Ethyl Formate and Their Combination, and the Sorption and Desorption of Fumigants on Cured Tobacco Leaves

**DOI:** 10.3390/insects11090599

**Published:** 2020-09-04

**Authors:** Bong Su Kim, Eun-Mi Shin, Young Ju Park, Jeong Oh Yang

**Affiliations:** Department of Plant Quarantine, Animal and Plant Quarantine Agency (APQA), Gimcheon 39660, Korea; bskim79@korea.kr (B.S.K.); eunmishin@korea.kr (E.-M.S.); bangjune@naver.com (Y.J.P.)

**Keywords:** ethyl formate, phosphine, combination, *Lasioderma serricorne*

## Abstract

**Simple Summary:**

*Lasioderma serricorne* (Fabricius) (Coleoptera: Anobiidae) is distributed throughout the world, where it is responsible for large amounts of economic damage to stored products in tropical and subtropical regions. To prevent the damage caused by this insect, the susceptibility of *L. serricorne* to phosphine (PH_3_), ethyl formate (EF), and their combination was evaluated in this study. Eggs, larvae, pupae and adults were subjected to treatment with fumigants to determine the 90% lethal concentration time values. The results show that, through treatment with PH_3_ + EF, control can be achieved at lower concentrations than for treatment with EF alone and at lower exposure times than for treatment with PH_3_ alone. The sorption rates of the fumigants on cured tobacco leaves were determined for the safety of workers, and EF required a ventilation time of longer than 22 h to desorb from cured tobacco leaves. Therefore, PH_3_ + EF can effectively control *L. serricorne* in cured tobacco leaves, with sufficient ventilation time required after treatment for the safety of workers.

**Abstract:**

The susceptibility of *Lasioderma serricorne* to phosphine (PH_3_), ethyl formate (EF) and their combination (PH_3_ + EF) was evaluated in this study. Eggs, larvae, pupae and adults were subjected to treatment with fumigants to determine the 90% lethal concentration time (LCt_90_) values. Treatment with PH_3_ for 20 h resulted in LCt_90_ values of 1.15, 1.39, 14.97 and 1.78 mg h/L while treatment with EF resulted in values of 157.96, 187.75, 126.06 and 83.10 mg h/L, respectively. By contrast, the combination of PH_3_ + EF resulted in LCt_90_ values of 36.05, 44.41, 187.17 and 35.12 mg h/L after 4 h. These results show that, through treatment with PH_3_ + EF, control can be achieved at lower concentrations than for treatment with EF alone and at lower exposure times than for treatment with PH_3_ alone. The sorption rates of the fumigants on cured tobacco leaves were determined for filling ratios of 2.5%, 5.0% and 10.0% (*w/v*). Cured tobacco leaves were treated with either 2 mg/L PH_3_, 114 mg/L EF or 0.5 mg/L PH_3_ + 109 mg/L EF. Treatment with PH_3_ showed sorption rates of 0.0%, 7.1% and 14.3%. EF, however, showed higher sorption rates of 64.9%, 68.5% and 75.5%, respectively, for the indicated filling ratios. When PH_3_ and EF were combined, the sorption rate of PH_3_ was 0.0%, while the sorption rates of EF were lower (9.1%, 12.0% and 23.2%) than treatment with only EF. EF required a ventilation time of longer than 22 h to desorb from cured tobacco leaves. Therefore, PH_3_ + EF can effectively control *L. serricorne* in cured tobacco leaves, with sufficient ventilation time required after treatment for the safety of workers.

## 1. Introduction

*Lasioderma serricorne* (Fabricius) (Coleoptera: Anobiidae) is distributed throughout the world, where it is responsible for large amounts of economic damage to stored products in tropical and subtropical regions [1,2]. *Lasioderma serricorne* is the insect considered to cause the most damage to tobacco, especially cured tobacco leaves, as well as cereals, dried fruits and cocoa beans [3,4]. *Lasioderma serricorne* damages tobacco by excavating tunnels and holes, and the incorporation of insect fragments and excreta into tobacco affects its taste, color and odor [5,6,7]. Generally, cured tobacco leaves are stored for many years before being made into cigarettes and are susceptible to loss resulting from inhabitation by *L. serricorne* [8]. To prevent the damage caused by this insect, the use of fumigants is recommended or even required, and they are specified in the law to include methyl bromide (MB) or phosphine (PH_3_) [9].

MB has been widely used because it rapidly kills insects, mites, microflora and nematodes [10]; however, it has been designated as an ozone-depleting substance by the Montreal Protocol, and its use is being phased out [11]. PH_3_ has the advantage of being safe for fresh commodities as it has lower phytotoxicity than other fumigants, but the disadvantage is that it requires a longer treatment period than other fumigants [12,13]. Ethyl formate (EF), an alternative to these fumigants, is one of the most widely used natural fumigants, and it is designated as “generally recognized as safe” by U.S. Food and Drug Administration [14,15,16,17]. In addition, co-treatment with PH_3_ and EF, rather than single treatment with an individual fumigant, has been reported to be effective against species or pests that are resistant to PH_3_ [17]. EF can achieve control when used at a high concentration over a short time, while PH_3_ can achieve control when used at a low concentration over a long time. Therefore, combination treatment with PH_3_ and EF (PH_3_ + EF) has been introduced to minimize the disadvantages while maximizing the advantages of each fumigant [17].

Fumigants should have a strong insecticidal effect without any phytotoxic effects on the stored food products [18]. In addition, they should not present lethal toxicity, nor should the use of fumigants affect the health of workers [19]. For this reason, the efficacy, phytotoxicity and worker safety of fumigants need to be considered together. This study aimed to investigate the activity of the PH_3_ and EF fumigants against eggs, larvae, pupae and adults of *L. serricorne* as part of a treatment strategy to prevent damage to cured tobacco leaves caused by *L. serricorne*. Both PH_3_ and EF were applied together to reinforce the effects of fumigation. In this study, cured tobacco leaves were treated, and the concentration of sorption was measured. The concentration of desorbed fumigant was measured to determine the appropriate time for fumigant release according to the threshold limit value (TLV).

## 2. Materials and Methods

### 2.1. Insects and Plants

Cigarette beetles (*L. serricorne*) were reared at FarmHannong Co., Ltd. (Chungcheongnam-do, Korea) and brought to the Plant Quarantine Technology Center (Gyeongsangbuk-do, Korea). The breeding conditions were maintained at 25 ± 1 °C and 60% ± 5% relative humidity in mixed flour, yeast and bran feed in a chamber (6.5 cm diameter × 18.0 cm high). Cured tobacco leaves were supplied by Korean Tobacco & Ginseng (Daejeon, Korea) and stored at 20 ± 1 °C and 50 ± 5% humidity. Tobacco leaves were cultivated in South Korea, and harvested leaves were fermented and dried in a warehouse owned by Korean Tobacco & Ginseng.

### 2.2. Fumigants

Phosphine gas (2% PH_3_ + 98% CO_2_) was purchased from Cytec (Sydney, NSW, Australia) as the ECO2Fume™ mixture. Ethyl formate (97%) was purchased from Aldrich Chemical Company Inc. (St. Louis, MO, USA).

### 2.3. Fumigation System

For fumigation bioassays of the fumigants, a desiccator (UBNC, Goyang, Korea) was loaded with petri dishes (dimensions 50 mm × 15 mm, ventilation hole size 13.2 mm, SPL, Seoul, Korea) containing mixed flour, yeast and bran feed inoculated with all stages (eggs, larvae, pupae and adults) of *L. serricorne*. Test insects were treated at PH_3_ 0, 0.025, 0.05, 0.1, 0.5, 1.0 and 1.5 mg/L for 20 h and EF 0, 10, 20, 30, 50 and 70 mg/L for 4 h in a 12 L desiccator. The PH_3_ + EF consisted of 0.5 mg/L PH_3_ and EF at 0, 5, 10, 15, 30, 50 and 80 mg/L for 4 h in a 55 L desiccator. Prior to injection of the fumigant, the dosage volume of air was removed from the desiccator using a gastight syringe (Hamilton, NV, USA) to avoid changes in pressure. All fumigation treatments were conducted at 20 ± 1 °C. The mortality of eggs was measured by counting the hatched eggs within 2 weeks after treatment as compared to the control. The mortality of larvae, pupae and adults were examined within 72 h after treatment.

### 2.4. Measurement of Fumigant Concentrations

To monitor the concentration of fumigants in the desiccator, gas samples were injected into Tedlar^®^ (SKC, Dorset, United Kingdom) bags (1 L) using a 50 mL syringe. PH_3_ was collected at 0.5, 1, 2, 4 and 20 h intervals, and EF and PH_3_ + EF were collected at 0.5, 1, 2 and 4 h intervals. The concentrations of PH_3_, EF and PH_3_ + EF in the desiccator were measured by gas chromatography (GC) analysis.

The PH_3_ concentration was measured on an Agilent GC 7890A equipped with an HP-PLOT/Q column (30 m × 530 μm × 40 μm, Agilent, Santa Clara, CA, USA) operated in split mode (10:1) and with a flame photometric detector. EF was measured using the Agilent GC 7890A equipped with a flame ionization detector (FID) after separation into split mode (10:1) in a Rtx-5 column (15 m × 250 μm × 1 μm, RESTEK). The injector and oven temperature were 200 °C. The detector temperature was 250 °C. The injection amount and flow rate of PH_3_ were 20 μL and 5 mL/min, and the injection amount and flow rate of EF were 70 μL and 1.5 mL/min. The concentrations of fumigants were calculated based on the peak area for the external standard.

### 2.5. Determination of the Concentration × Time (Ct) of the Fumigants

The efficacy of a fumigant can be affected by the concentration and fumigation time [20]. For this reason, the effect of the fumigant on insects and the commodity was expressed as the concentration × time (Ct) product. The Ct value of fumigants was calculated using the equation outlined by Monro [21]. The fumigants were monitored in terms of concentration values at timed intervals over exposure time through GC analysis.

### 2.6. Evaluation of the Sorption and Desorption of Fumigants on Cured Tobacco Leaves

Cured tobacco leaves (300, 600 and 1200 g) were placed at ratios of 2.5%, 5.0% and 10.0% (*w/v*) in a 12 L desiccator and treated with either 2 mg/L PH_3_, 114 mg/L EF or 0.5 mg/L PH_3_ + 109 mg/L EF. PH_3_ was used for 20 h, while EF and PH3 + EF were used for 4 h (20 ± 1 °C). Empty desiccators without tobacco leaves were used as a negative control to compare the sorption of the fumigants. Before injecting the fumigant into the desiccator, a dosage volume of air was removed using a gastight syringe to avoid changes in pressure. The concentration of PH_3_ was measured by GC analysis at 0.5, 2, 4 and 20 h after treatment, and EF and PH_3_ + EF were measured at 0.5, 1, 3 and 4 h after treatment. The concentration of adsorbed gas was calculated according to Ren et al. [22] and Lee et al. [23]. The fumigant adsorbed in treated tobacco leaves was ventilated for 0.5, 2, 4 and 24 h, and the treated tobacco leaves (300 g) were resealed in 3 L desiccators to measure the desorption rate. After storing the tobacco leaves in 3 L desiccators for 6 h at room temperature, the concentration of fumigant inside the desiccator was measured by GC.

### 2.7. Statistical Analysis

All treatments of fumigants were performed using three replicates. The mean of mortality and standard error (SE) of *L. serricorne* were calculated using Microsoft Excel 2013. The lethal concentration time values for 50% and 90% mortality (LCt_50_ and LCt_90_ values) were calculated using probit analysis [24]. The treatment values were compared and analyzed using Tukey’s test at *p* < 0.05 (SPSS Inc., Chicago, IL, USA, 2009).

## 3. Results

### 3.1. Efficacy of Fumigants Against L. Serricorne

Mortality of pupae was lower compared to eggs, larvae and adults when PH_3_ was treated with less than 1 mg/L (Figure 1). The LCt_50_ values for PH_3_ were highest in the pupae (3.75 mg h/L), followed by adults (0.70 mg h/L), larvae (0.65 mg h/L) and eggs (0.32 mg h/L) (Table 1). The LCt_90_ values were highest for pupae (14.97 mg∙h/L), followed by adults (1.78 mg∙h/L), larvae (1.39 mg∙h/L) and then eggs (1.15 mg∙h/L).

EF showed 100% mortality at 70 mg/L against *L. serricorne* eggs, pupae and adults (Figure 2). At this concentration of EF, the highest LCt_50_ value of 137.61 mg h/L was observed for *L. serricorne* larvae, 72.14 mg h/L for the pupae, 52.95 mg h/L for the adults and 42.66 mg h/L for the eggs, as determined by probit analysis with high tolerance. The overall highest observed LCt_90_ values were 187.75 mg h/L for larvae, followed by 157.96 mg h/L for eggs, 126.06 mg h/L for pupae and 83.10 mg h/L for adults (Table 2).

The combination treatment of PH_3_ (0.5 mg/L) + EF showed 100% mortality at an EF concentration of 30 mg/L for the eggs and adults, 50 mg/L for the larvae and 80 mg/L for the pupae (Figure 3). The highest values of LCt_50_ were 40.43 mg h/L for the pupae, followed by 15.84 mg h/L for eggs, 13.13 mg h/L for adults and 9.89 mg h/L for larvae (Table 3). The overall highest observed LCt_90_ values were 187.17 mg h/L for pupae, followed by 44.41 mg h/L for larvae, 36.05 mg h/L for eggs and 35.12 mg h/L for adults.

### 3.2. Sorption and Desorption Rate of Fumigants on Cured Tobacco Leaves

The sorption rates of PH_3_, EF and PH_3_ + EF fumigants on cured tobacco leaves were investigated by administering the fumigants at volume ratios of 2.5%, 5.0% and 10.0% (*w/v*) in a 12 L desiccator. At volume ratios of 2.5%, 5.0% and 10.0% (*w/v*), PH_3_ exhibited sorption rates of 0.0%, 7.1% and 14.3% (Figure 4) and EF rates of 64.9%, 68.5% and 75.5%, respectively (Figure 5). The combination treatment (PH_3_ + EF) showed a PH_3_ sorption rate of 0.0% and EF rate of 9.1%, 12.0% and 23.2%, respectively (Figure 6).

The treated tobacco leaves were resealed after 0.5, 2, 4 and 24 h of ventilation in the 12 L desiccator, and the remaining concentration was measured to determine the appropriate time for fumigant release according to the TLV. PH_3_ required more than 0.5 h ventilation to fall below TLV (0.0004 mg/L) (Figure 7), whereas the concentration of EF in the cured tobacco leaves decreased after the end of exposure but did not fall below TLV (0.3 mg/L), and a ventilation time of approximately 22 h was required to reach the TLV (Figure 8). For the combination treatment (PH_3_ + EF), PH_3_ required approximately 0.5 h, whereas EF required longer than 24 h (Figure 9). For worker safety, EF-treated cured tobacco leaves require a ventilation time of 22 h or more.

## 4. Discussion

A number of chemicals (phosphine, sulfuryl fluoride, carbonyl sulfide, carbon dioxide, carbon disulfide, ethyl formate, ethylene oxide, hydrogen cyanide, methyl iodide, methyl isothiocyanate, ozone, sulfur dioxide, ethyl or methyl formate and acetaldehyde) are considered alternative fumigants to MB but do not appear to be as effective [10,25,26,27]. Some have been found to cause phytotoxic effects in the fumigated products, exhibit limited penetration, leave residues or are otherwise unsuitable for health or economic reasons [26,27,28].

This study confirmed the insecticidal effects of PH_3_ and EF against *L. serricorne*. PH_3_ differs from other fumigants in that long exposures at lower concentrations are more effective than higher concentrations [29,30,31]. The treatment of *L. serricorne* with PH_3_ at a concentration of 1.0 mg/L for 20 h showed an excellent mortality rate of 86.36% or more, but a long exposure time was required. The characteristics of PH_3_ are supported by studies on *Tetranychus urticae* eggs. *T. urticae* eggs showed a hatch rate of 98.0% after 6 h and 9.5% after 24 h following treatment with PH_3_ 1.0 mg/L [23]. This study showed that 70 mg/L for 4 h EF treatment resulted in a mortality rate of more than 98% for all stages of *L. serricorne*. Lee et al. [23] reported a high mortality rate of 62.4% for *Sitophilus oryzae* (mixed age) treated with 67.4 mg/L EF for 6 h. These high concentrations render EF are economically disadvantageous for use in the field. However, studies examining the combination of PH_3_ + EF against *Aphis gossypii*, *Lipaphis erysimi*, *Myzus persicae*, *Planococcus citri* and *T. urticae* showed that the combined treatment required reduced fumigant concentrations and treatment times compared to individual fumigant treatment [17,32,33,34,35]. This study investigated the insecticidal effect of PH_3_ + EF against *L. serricorne* using a range of EF concentrations, and treatment at 50 mg/L EF for 4 h showed a mortality rate of 88% or more for all stages. This is a lower concentration than required for EF alone to produce a similar effect and, also, a shorter treatment time than for use of PH_3_ alone. Yang et al. [17] showed that a lower concentration and shorter treatment time were required for combined PH_3_ + EF treatment of *P. citri*. PH_3_ and EF treatment have the same mode of action, so more studies are required for combined PH_3_ + EF treatment to analyze the cause of excellent fumigation effects against *L. serricorne* at low concentrations and short treatment times [36,37].

Another important feature of fumigants is low sorption onto commodities. When fumigants exhibit high sorption on commodities, their efficacy is lowered because of the decreased concentration of fumigants in the air, which weakens their effects [38,39,40]. Treatment with 84.4 mg h/L EF or higher resulted in phytotoxicity toward asparagus, with no fumigation effect against *F. occidentalis* eggs [40]. The sorption rates of PH_3_, EF and PH_3_ + EF on cured tobacco leaves were found to increase with the volume ratio. The highest sorption rate of EF (75.5%) was obtained at a filling ratio of 10.0% (*w/v*). On the other hand, the PH_3_ + EF treatment showed a low sorption rate of 0.03% for PH_3_ and 23.2% for EF, and the efficacy of the fumigant was increased due to the lower sorption rate.

The TLV indicates the level of exposure a worker can tolerate in a day, assuming a lifetime of working without adverse effects. Worker safety should be considered by confirming the appropriate time for expulsion of the fumigant to the TLV to minimize exposure. PH_3_, EF and PH_3_ + EF were used to treat cured tobacco leaves, and the TLVs of each fumigant were compared with the concentration of the fumigant remaining in the commodities based on the time of evacuation. PH_3_ reached the TLV within 0.5 h after evacuation, while for EF, more than 22 h was required. EF shows rapid sorption and degradation in high-temperature or humid commodities [15,41,42,43,44]. Although the concentration of fumigant depends on the temperature and humidity, this study shows that, in order to ensure that the fumigant has reached the TLV, workers should resume work only after a sufficient time period has elapsed.

## 5. Conclusions

In this study, we attempted to control *L. serricorne* using PH_3_ and EF as fumigants to replace MB. Although both can achieve effective control, PH_3_ requires a long exposure time for fumigation, whereas large amounts of EF are required. As an alternative, *L. serricorne* was treated with a combination of PH_3_ and EF. This combined fumigant was used at a concentration similar to that of MB, resulting in shorter exposure times to achieve comparable effects. Therefore, PH_3_ and EF can be effectively used to control *L. serricorne* in cured tobacco leaves, but prudent application is required to ensure worker safety.

## Figures and Tables

**Figure 1 insects-11-00599-f001:**
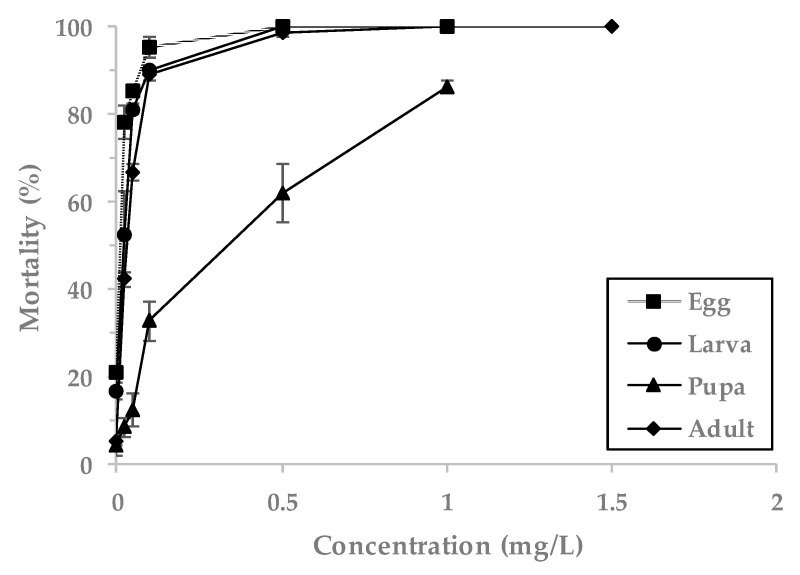
Mortality of *L. serricorne* exposed to the PH_3_ fumigant for 20 h at 20 °C in a 12 L desiccator.

**Figure 2 insects-11-00599-f002:**
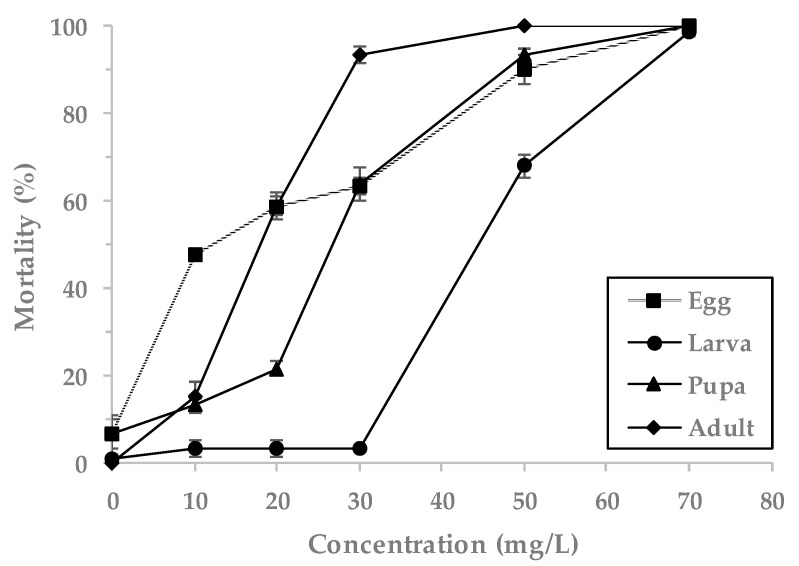
Mortality of *L. serricorne* exposed to the EF fumigant at indicated concentrations for 4 h at 20 °C in a 12 L desiccator.

**Figure 3 insects-11-00599-f003:**
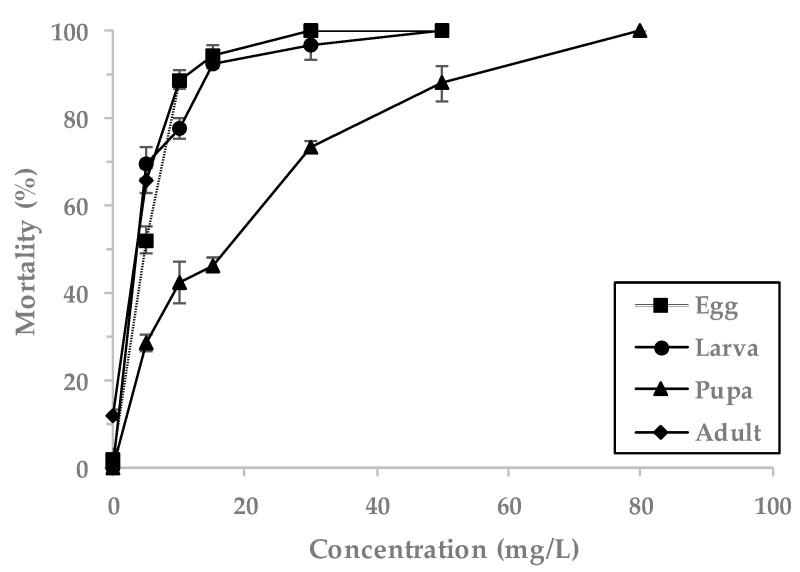
Mortality of *L. serricorne* exposed to gas mixtures of PH_3_ (0.5 mg/L) + EF at indicated concentrations for 4 h at 20 °C in a 55 L desiccator.

**Figure 4 insects-11-00599-f004:**
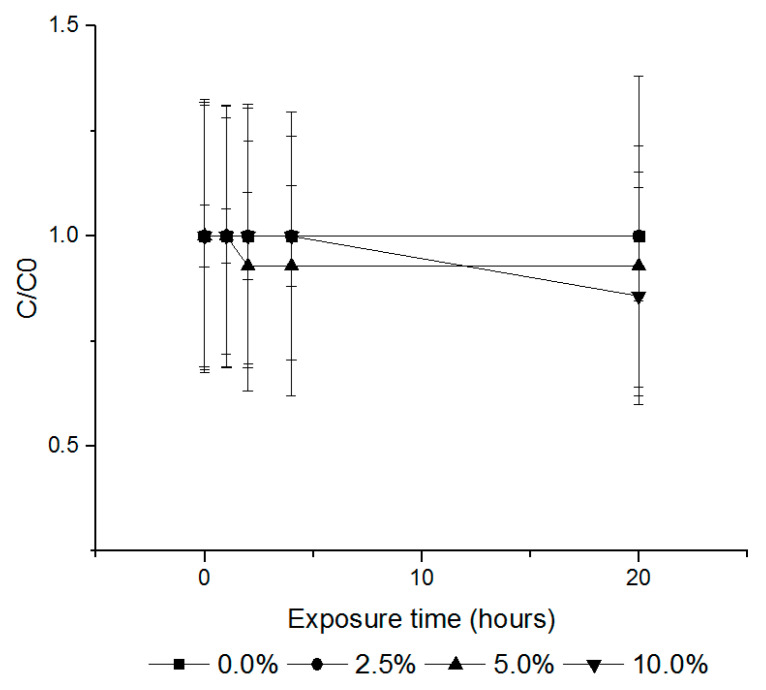
The loss amounts of PH_3_ according to the different loading ratios of cured tobacco leaves (control, 2.5%, 5.0% and 10.0% (*w/v*)) during fumigation for 20 h (20 ± 1 °C). Each value represents the mean ± SE (*n* = 3). C/C0 is the ratio of the concentration of the fumigant in the headspace.

**Figure 5 insects-11-00599-f005:**
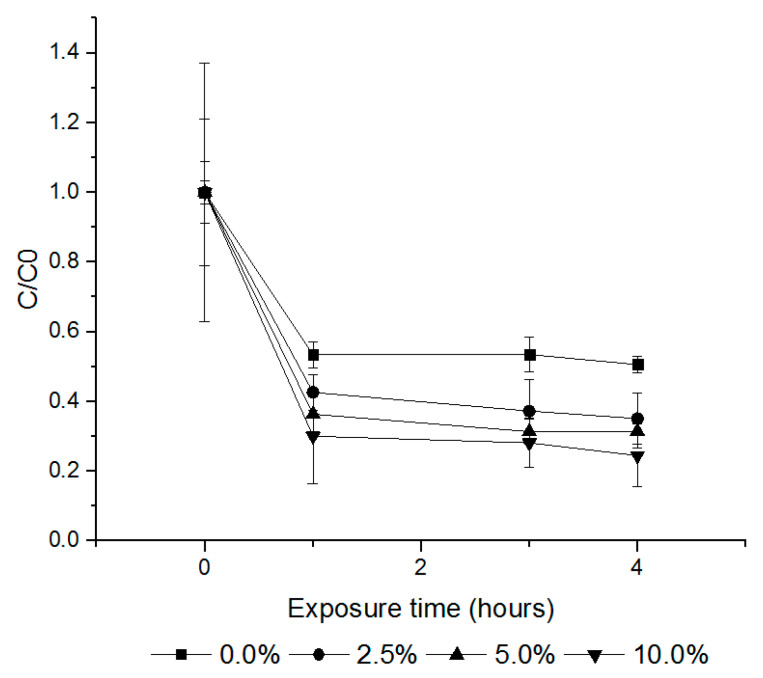
The amount of EF loss according to the different loading ratios of cured tobacco leaves (control, 2.5%, 5.0% and 10.0% (*w/v*)). Each value represents the mean ± SE (*n* = 3). C/C0 is the ratio of the concentration of the fumigant in the headspace.

**Figure 6 insects-11-00599-f006:**
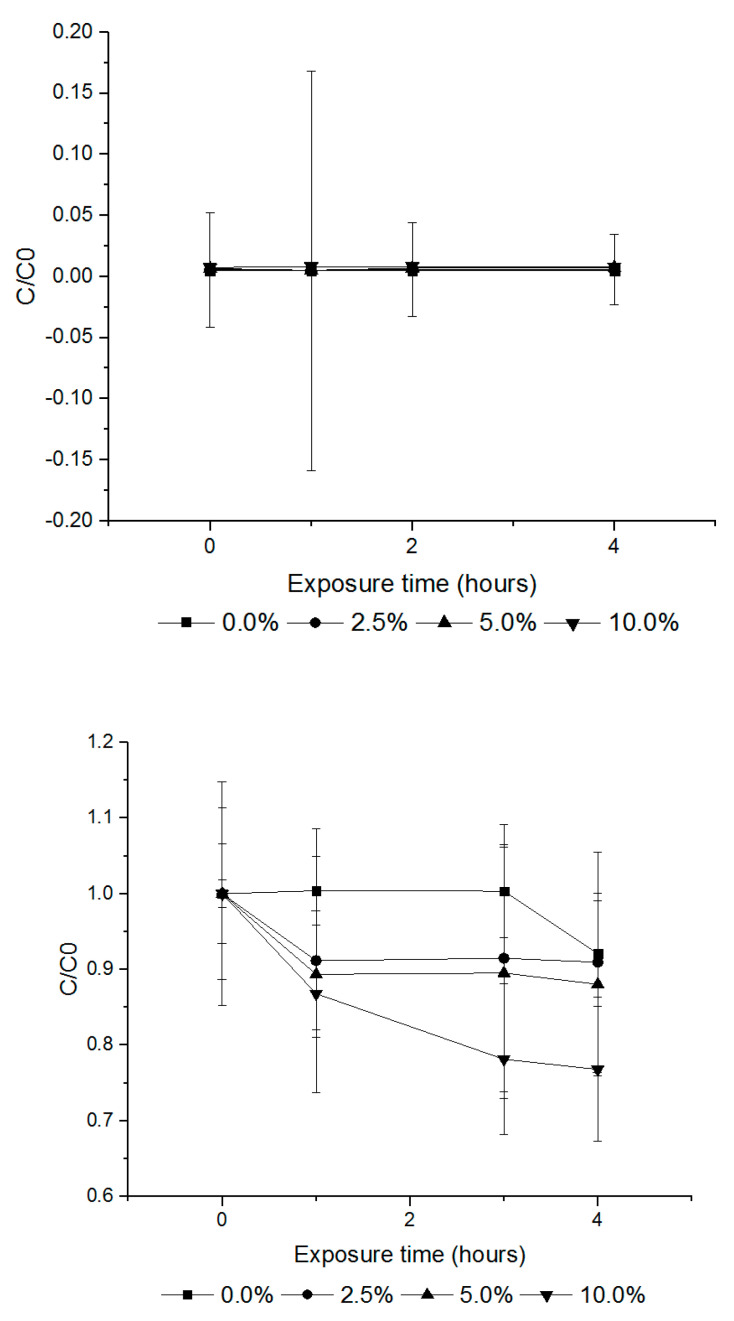
The amount of PH_3_ + EF loss according to the different loading ratios of cured tobacco leaves (control, 2.5%, 5.0% and 10.0% (*w/v*)). Each value represents the mean ± SE (*n* = 3). C/C0 is the ratio of the concentration of the fumigant in the headspace.

**Figure 7 insects-11-00599-f007:**
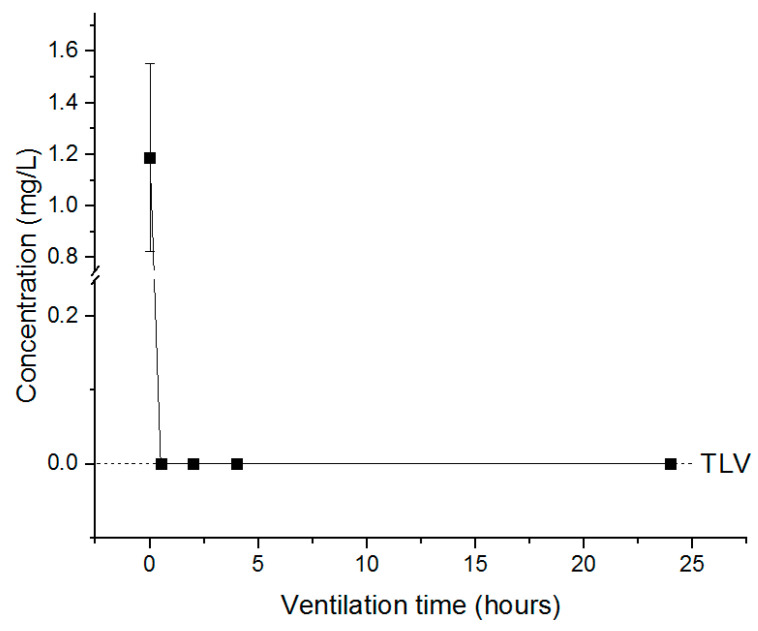
The concentration of PH_3_ inside the desiccator under resealed conditions. Each value represents the mean ± SE (*n* = 3).

**Figure 8 insects-11-00599-f008:**
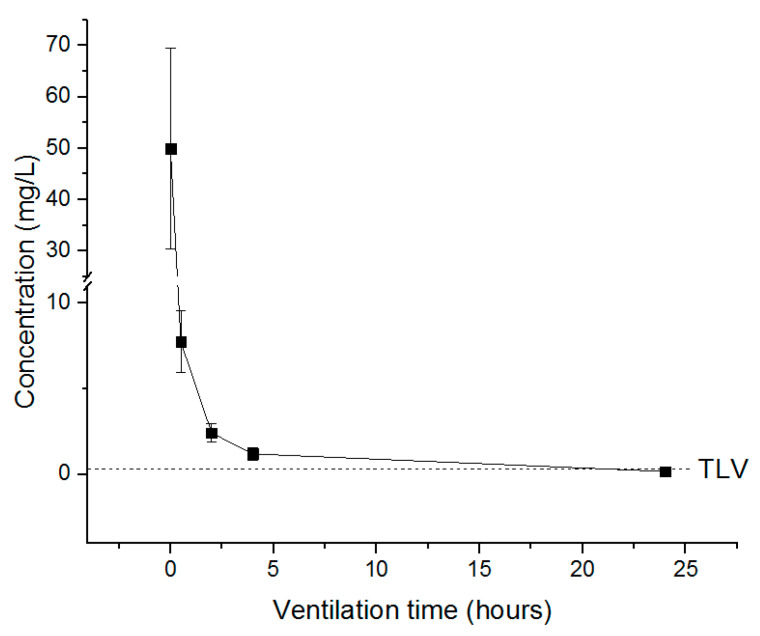
The concentration of EF inside the desiccator under resealed conditions. Each value represents the mean ± SE (*n* = 3).

**Figure 9 insects-11-00599-f009:**
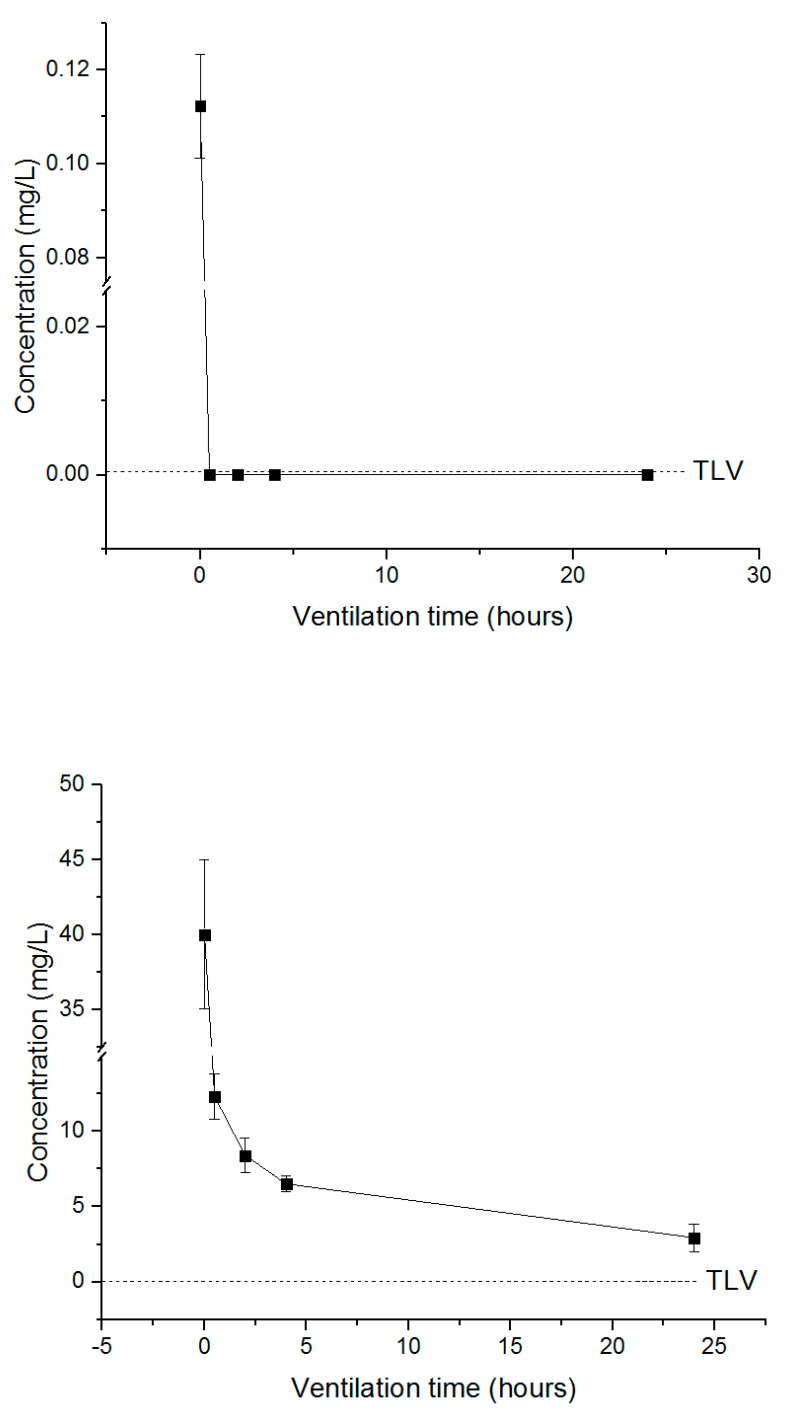
The concentration of PH_3_ + EF inside the desiccator under resealed conditions. Each value represents the mean ± SE (*n* = 3).

**Table 1 insects-11-00599-t001:** Lethal concentration time of *L. serricorne* exposed to the PH_3_ fumigant.

Stages	*n*	LCt_50_ (mg h/L) (95% CL ^a^)	LCt_90_ (mg h/L) (95% CL)	Slope ± SE	*df*	χ^2^
Egg	426	0.32 (0.00–0.72)	1.15 (0.20–1.68)	2.29 ± 0.56	5	2.87
Larva	543	0.65 (0.11–1.06)	1.39 (0.65–1.78)	3.85 ± 0.70	5	5.86
Pupa	290	3.75 (2.59–6.39)	14.97 (8.16–53.08)	2.13 ± 0.26	5	2.94
Adult	632	0.70 (0.00–1.43)	1.78 (0.25–2.56)	3.14 ± 0.83	6	2.18

^a^ CL denotes the confidence limit.

**Table 2 insects-11-00599-t002:** Lethal concentration time of *L. serricorne* exposed to the EF fumigant.

Stages	*n*	LCt_50_ (mg h/L) (95% CL ^a^)	LCt_90_ (mg h/L) (95% CL)	Slope ± SE	*df*	χ^2^
Egg	535	42.66 (4.12–71.57)	157.96 (94.87–1394.50)	2.25 ± 0.54	5	15.68
Larva	541	137.61 (110.51–602.10)	187.75 (134.79–2672.46)	9.50 ± 2.37	5	0.36
Pupa	361	72.14 (55.33–93.18)	126.06 (96.63–2477.82)	5.29 ± 0.87	5	6.40
Adult	540	52.95 (39.85–62.41)	83.10 (72.91–92.07)	6.55 ± 0.63	5	0.00

^a^ CL denotes the confidence limit.

**Table 3 insects-11-00599-t003:** Lethal concentration time of *L. serricorne* exposed to EF fumigant mixed with PH_3_ (0.5 mg/L).

Stages	*n*	LCt_50_ (mg h/L) (95% CL ^a^)	LCt_90_ (mg h/L) (95% CL)	Slope ± SE	*df*	χ^2^
Egg	540	15.84 (6.57–23.38)	36.05 (25.02–44.39)	3.58 ± 0.46	5	0.05
Larva	545	9.89 (1.32–18.86)	44.41 (27.13–64.98)	1.92 ± 0.36	5	3.81
Pupa	445	40.43 (23.97–56.84)	187.17 (123.60–414.96)	1.75 ± 0.21	6	2.92
Adult	541	13.13 (3.89–21.33)	35.12 (21.80–45.27)	3.01 ± 0.45	5	1.32

^a^ CL denotes the confidence limit.

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
