# Peer review of "Susceptibility of the Cigarette Beetle Lasioderma serricorne (Fabricius) to Phosphine, Ethyl Formate and Their Combination, and the Sorption and Desorption of Fumigants on Cured Tobacco Leaves"

_insects, 2020, doi:10.3390/insects11090599_

Round 1

Reviewer 1 Report

  • the paper is very interesting 
  • The author should write some information about Tobacco plant
  • This sentence “Cured tobacco leaves (300, 600, and 1200 g) were placed at ratios of 2.5%, 5.0%, and 10.0% (w/v) 120 in a 12 L desiccator and treated with either 2 mg/L PH3, 114 mg/L EF, or 0.5 mg/L PH3 + 109 mg/L EF. PH3 was used for 20 h, while EF and PH3 + EF were used for 4 h (20 ± 1 °C).” is not clear.

Author Response

The author should write some information about Tobacco plant;

-> Added information about manufacturing process of tobacco plant that used for this experiment (80~83)

“Cured tobacco leaves were supplied by Korean Tobacco & Ginseng (Daejeon, South Korea) and stored at 20 ± 1 °C and 50 ± 5% humidity. Tobacco leaves were cultivated in South Korea, and harvested leaves were fermented and dried in a warehouse owned by Korean Tobacco & Ginseng.”

The cigarette beetle only exists in stored dried leaves, not in a fresh plant, so we omitted the information of fresh tobacco plant.

This sentence “Cured tobacco leaves (300, 600, and 1200 g) were placed at ratios of 2.5%, 5.0%, and 10.0% (w/v) 120 in a 12 L desiccator and treated with either 2 mg/L PH3, 114 mg/L EF, or 0.5 mg/L PH3 + 109 mg/L EF. PH3 was used for 20 h, while EF and PH3 + EF were used for 4 h (20 ± 1 ℃).” is not clear

-> This sentence means the filling ratios of leaves are 2.5% (300g / 12L), 5.0% (600g / 12L) and 10.0% (1,200g / 12L) (w/v), and the three dosage (include EF, PH3 and EF+PH3) were treated to all filling ratio (3 x 3 treatments).

Reviewer 2 Report

This study excels because it takes into account two aspects of applicability: insecticide efficacy and human exposure.

Specific comments by line number:

51: Delete 'either'.

58: Explain 'GRAS' or leave out.

141-142: You should not report differences that are not statistically significant. The pupae differed from the other life stages; the other life stages were alike. You could write, for instance: 'Pupae had the highest LCt50 (3.75 mg h/L) compared to the other life stages (estimated at 0.32 to 0.70 mg h/L)'. So, don't present life stages in falling order of LCt50 when, in fact, there values do not differ statistically. Report your findings in the same, more conservative manner, thoughout you Results section.

Author Response

51: Delete ‘either’

-> Deleted (50~51)

58: Explain ‘GRAS’ or leave out

-> Modified the sentence (57~59)

“Ethyl formate (EF), an alternative to these fumigants, is one of the most widely used natural fumigants, and it is designated as “generally recognized as safe” by U.S. Food and Drug Administration”

141-142: You should not report differences that are not statistically significant. The pupae differed from the other life stages; the other life stages were alike. You could write, for instance: ‘Pupae had the highest LCt50 (3.75 mg h/L) compared to the other life stages (estimated at 0.32 to 0.70 mg h/L)’. So don’t present life stages in falling order of LCt50 when, in fact, there values don not differ statistically. Report your finding in the same, more conservative manner, throughout you Results section.

-> Modified as below:

“Mortality of pupae was lower compared to eggs, larvae, and adults when PH3 was treated with less than 1 mg/L (Figure 1).”

Reviewer 3 Report

Susceptibility of the Cigarette Beetle Lasioderma serricorne (Fabricius) to Phosphine, Ethyl Formate and Their Combination, and the Sorption and Desorption of Fumigants on Cured Tobacco Leaves

The manuscript is written very clearly and logically. The methods and conclusions are sound. Stored product pests cause significant economic losses each year around the world. Safe and effective control measures are needed, and this paper describes work to examine the efficacy and safety of a combination of two fumigants. The results contribute to our knowledge of stored product management and should be of commercial value to the tobacco industry. While I am not an expert on stored product pest IPM, my assessment is that this paper is suitable for publication with only minor revisions.

Line 35: ratios not “rations”

Line 45: Genus names are generally spelled out when they appear as the first word in a sentence.

Line 57: delete “…plants treated..”

Lines 110 and 111: the injection amount was “20 μm” and “70 μm”; I was expecting a unit of volume rather than microns.

Line 125: The concentration of PH3 was measured…

Line 220: “…of PH3 sre supported by studies on Tetranychus urticae eggs.”

Line 222: Why is no unit of time given?  “…70 mg/L EF treatment resulted…”

Line 224: Reword the following: These high concentrations render EF is economically disadvantageous for use in the field.

Author Response

Line 35: ratio not “rations”

-> fixed

Line 45: Genus names are generally spelled out when they appear as the first word in a sentence.

-> fixed (both 45 and 47)

Line 57: delete “…plants treated…”.

-> deleted

Line 110 and 111: the injection amount was “20um” and “70um”; I was expecting a unit of volume rather than microns.

-> fixed as “20 uL” and “70 uL”

Line 125: The concentration of PH3 was measured…

-> fixed

Line 220: “…of PH3 are supported by studies on Tetranychus urticae eggs”

-> fixed

Line 222: Why is no unit of time given? “70 mg/L EF treatment resulted”

-> fixed as “70 mg/L for 4 h EF treatment resulted”

Round 2

Reviewer 1 Report

i suggest to accept the article in the current form